# Laparoscopic versus Robotic Hepatectomy: A Systematic Review and Meta-Analysis

**DOI:** 10.3390/jcm11195831

**Published:** 2022-09-30

**Authors:** Taslim Aboudou, Meixuan Li, Zeliang Zhang, Zhengfeng Wang, Yanfei Li, Lufang Feng, Xiajing Chu, Nan Chen, Wence Zhou, Kehu Yang

**Affiliations:** 1The First School of Clinical Medicine, Lanzhou University, Lanzhou 730000, China; 2The First Hospital of Lanzhou University, Lanzhou University, Lanzhou 730000, China; 3Evidence Based Medicine Center, School of Basic Medical Sciences, Lanzhou University, Lanzhou 730000, China; 4Key Laboratory of Evidence Based Medicine and Knowledge Translation of Gansu Province, Lanzhou 730000, China

**Keywords:** meta-analysis, hepatic, robotic, outcomes, laparoscopic

## Abstract

This study aimed to assess the surgical outcomes of robotic compared to laparoscopic hepatectomy, with a special focus on the meta-analysis method. Original studies were collected from three Chinese databases, PubMed, EMBASE, and Cochrane Library databases. Our systematic review was conducted on 682 patients with robotic liver resection, and 1101 patients were operated by laparoscopic platform. Robotic surgery has a long surgical duration (MD = 43.99, 95% CI: 23.45–64.53, *p* = 0.0001), while there is no significant difference in length of hospital stay (MD = 0.10, 95% CI: −0.38–0.58, *p* = 0.69), blood loss (MD = −20, 95% CI: −64.90–23.34, *p* = 0.36), the incidence of conversion (OR = 0.84, 95% CI: 0.41–1.69, *p* = 0.62), and tumor size (MD = 0.30, 95% CI: −0–0.60, *p* = 0.05); the subgroup analysis of major and minor hepatectomy on operation time is (MD = −7.08, 95% CI: −15.22–0.07, *p* = 0.09) and (MD = 39.87, 95% CI: −1.70–81.44, *p* = 0.06), respectively. However, despite the deficiencies of robotic hepatectomy in terms of extended operation time compared to laparoscopic hepatectomy, robotic hepatectomy is still effective and equivalent to laparoscopic hepatectomy in outcomes. Scientific evaluation and research on one portion of the liver may produce more efficacity and more precise results. Therefore, more clinical trials are needed to evaluate the clinical outcomes of robotic compared to laparoscopic hepatectomy.

## 1. Introduction

In addition to other variables, the surgical method decided on by surgeons is an important factor capable of swaying the outcomes of hepatectomies. Laparoscopic liver resection has increasingly been proposed as a possible appropriate treatment for patients with hepatocellular carcinoma, especially for tumors in the anterior segments [1,2,3]. Having been considered suitable in diverse scenarios based on feasibility and efficacy, laparoscopy has since gained a significant reputation as a choice approach for liver resection in many parts of the world [3,4]. Many studies have demonstrated the efficacy of laparoscopic platforms compared to traditional open surgery in the repair of minor bleeding, hospital duration, and improved cosmetic recovery [5,6,7]. In terms of perioperative results, several studies have shown that laparoscopic liver resection might be better than open liver resection, especially in patients with cirrhosis [7,8,9]. A lower postoperative occurrence of ascites is reported when laparoscopic liver resection is used. Furthermore, since laparoscopic liver resection minimizes the interruption of the portosystemic collateral vessels because the incisions in the anterior abdominal wall are small, the rate of liver failure and the recurrence of ascites after this procedure in patients with severe cirrhosis are reduced [10,11,12,13].

The Barcelona Clinic Liver Cancer (BCLC) system is an incorporated hepatocellular carcinoma staging device used commonly worldwide. It associates a patient diagnostic with an evidence-based treatment opportunity at any evolutionary stage, including the function of the liver, physical status, and tumor extension. In the BCLC staging system, the Child–Pugh–Turcotte (CPT) classification is used to evaluate the function of the liver. It is based on a score derived from five parameters: bilirubin and albumin serum concentration, prothrombin time, and the existence and complication of ascites and hepatic encephalopathy [14,15,16,17]. Some surgeons believe that laparoscopic liver resection can be safely used in CPT B patients and does not cause substantial postoperative complications, such as intractable ascites [16]. The fear that laparoscopic liver resection could reduce the surgical margin because palpation is not possible might be counterbalanced by the systematic use of intraoperative ultrasound, making it possible to maintain the intended margin. Five-year overall survival and recurrence of laparoscopic liver resection [16,18] are similar to those in other studies using conventional surgical resection [14,19]. Additionally, [7,8,9,10,11] emphasized the above-mentioned advantages associated with laparoscopy and also highlighted some obvious shortcomings of the procedure that require further research and innovation to overcome. Such research would be required to clarify the role of laparoscopic liver resection particularly in patients with CPT B cirrhosis, including prospective randomized controlled trials. It is, however, difficult to identify acceptable inclusion criteria on which all surgeons can agree. 

Despite the characteristic limitations of laparoscopic equipment in terms of restrictions and challenges for surgical liver resection, i.e., the seven degrees of freedom of laparoscopic instruments and the two-dimensional view, poorer ergonomics are seen primarily through the extended nature of the procedure, hand tremor, and surgeon fatigue [9]. These apparent effects remain the most important interruption to its broader application in complex abdominal surgical platforms [20]. Recently, minimally invasive surgical techniques have emerged opening new perspectives for the surgical treatment of patients with hepatocellular carcinoma, including extended right and left hepatectomies. One such technique is the application of robotic-assisted computer surgery, which improves on the limitations of traditional laparoscopic surgery. First introduced in the 1990s, this is conducted by three-dimensional (3D) visualization and instruments with seven degrees of freedom. In addition, occurrence of hand tremor is less intense and surgeon siting position during the entire time period of the operation [21,22] is improved, contributing to an increased consideration of this approach. Therefore, more robotic hepatectomies have been performed due to the improvement of surgical instruments, while many research papers have been documented for comparative purposes.

Several reviews compare the outcomes of robotic and laparoscopic liver resections, but without meta-analysis. Surgeons agree on the fact that significant achievements have been made in both robotic methods and laparoscopy; however, it remains a daunting challenge to decide which is better for liver surgery. The present study explores recent progress in this area of research with focus on studies that compare robotic platforms with laparoscopic techniques during hepatectomy procedures, and makes deep comparisons using meta-analysis for clearer observations on efficacy and safety.

## 2. Methods

### 2.1. Literature Search

A literature quest was conducted on PubMed, Embase, Cochrane Library, and three Chinese databases through the following search strategies to segregate studies (Figure 1) based on the inclusion and exclusion criteria (Section 2.2). The search strings were as follows: (“hepato [Title/Abstract]” OR “liver [Title/Abstract]” OR “liver [Mesh]”) AND (“Resection [Title/Abstract] OR “resections [Title/Abstract]” OR “segmentectomy [Title/Abstract]”) OR “segmentectomies [Title/Abstract]”) AND (“Robotic Surgical Procedures [Title/Abstract]” “Robotics [Title/Abstract]” OR “Robotics [Mesh]”). In addition, we checked the references of any related review articles or meta-analysis to find more eligible studies and all our research was performed in the English language.

### 2.2. Inclusion and Exclusion Criteria

Studies were included if they met the flowing criteria: (a) population: patients diagnosed with liver cancer; (b) intervention: robotic liver resection versus laparoscopic surgery. (c) outcomes: no restriction; (d) study design: no restriction. The exclusion criteria were as follows: (a) duplicate reports of a study; (b) studies with insufficient data and without the author’s response (e.g., protocols, conference proceedings or abstracts, among others). 

### 2.3. Study Selection and Data Extraction

The screening and extraction of data were conducted separately by two independent reviewers [23]. In opposing views between the two reviewers, a third reviewer [24,25] was invited to reconcile the differences. Duplicate articles were detected and removed using EndNote X8 software (Thomson Corporation; Stamford, CT, USA). Subsequently, the reviewers screened the titles and abstracts of the selected articles. An article was denied further review when both reviewers excluded it. Article full text was obtained and examined for suitability when one reviewer only included it, or when the title and abstract did not provide sufficient information to make a decision. General data information about the year of publication, the author’s first name, trial design, sample size, as well as the patient’s characteristics, such as gender, type of disease and mean age, were extracted into a predesigned table. The details of the intervention, including the duration and treatment techniques and risk of bias and outcomes data were also extracted.

### 2.4. Publication Bias

The funnel plot of this study created on complication rates is shown in (Figure 2). Inside the limits of the 95 % CIs and distributed more evenly about the vertical, it suggests there is no publication bias because of pot symmetry.

### 2.5. Data Analysis

RevMan version 5.3 (Copenhagen: The Nordic Cochrane Center, The Cochrane Collaboration, London, UK) was used in conducting the meta-analysis. Variables that were dichotomous were assessed by the use of risk ratio (OR) at a confidence interval of 95% (95% CIs). Mean differences (MDs) were used in analyzing variables that were continuous, also at 95% CIs. Statistical algorithms were employed to generate the precise means and standards of continuous variables from studies that were presented in *p* values, ranges, and medians. To pool the studies, a random effect model was used. The Mantel–Haenszel method was used to conduct the meta-analysis on binary variables, while the inverse variance method was used for the continuous variables. The evaluation of heterogeneity was performed using I^2^ statistics and the Cochran Q test. Studies of low quality were excluded by sensitivity test.

### 2.6. Study Selection

A flow diagram of the literature selection process is presented in (Figure 1). A total of 2320 studies relevant to the search terms were retrieved; 66 of these were excluded based on duplication. Two thousand two hundred fifty-four records were excluded based on screening of titles or abstracts, of which 2206 were considered not eligible. The full texts of the remaining 48 articles were screened for a more accurate estimate, and 29 trials were excluded from our analysis. Finally, 19 studies met our inclusion criteria.

### 2.7. Study Characteristics

Nineteen studies [26,27,28,29,30,31,32,33,34,35,36,37,38,39,40,41,42,43,44] elucidated the outcomes of robotic and laparoscopic surgery, with 682 and 1101 patients, respectively. The characteristics of studies included in the meta-analysis are shown in (Table 1 and Table 2). All included studies were published between 2010–2021. Of the included trials, six trials (31.57%) were conducted in China, four in the United States (21.05%), with three studies in South Korea (15.8%), two in Germany (10.52%), one for Italy, Russia, Belgium-Italy, and France-Italy (5.26%) for each country and collaboration, respectively [28,29,30,31,32,33,34,35,36,37,38,39,40,41,42,43,44,45,46]. Furthermore, for the characterisation of major hepatectomy resection of 3 Couinaud liver segments, additional hemihepatectomy was included in the analysis of the studies with major liver resection, as demonstrated in (Table 1 and Table 2). Moreover, in the subgroup of minor liver resection, we included studies such as right or left lateral hepatectomy (Table 2).

#### The Results of Meta-Analyses

Blood transfusion rate

We included ten studies [26,29,31,32,34,36,37,39,42,43], of 825 patients evaluating the blood transfusion rate of robotic surgery for Hepatectomy. Our meta-analysis results demonstrated that there was no significant difference between robotic and laparoscopic surgery in reducing the blood transfusion rate of hepatectomy patients (OR = 1.33, 95%CI: 0.69–2.56, *p* = 0.39). The heterogeneity was observed with low certainty, I^2^ = 13%, *p* = 0.33, presented in Figure 3. 

Complications

Eleven trials [26,27,29,30,32,33,34,35,36,37,43], including 1206 patients, evaluated the outcomes of complications with patients undergoing robotic hepatectomy. Our meta-analysis results revealed that the difference between robotic and laparoscopic surgery in decreasing complications with patients undergoing hepatectomy was insignificant (OR = 0.94, 95%CI: 0.66–1.35, *p* = 0.75). The heterogeneity was observed with low certainty I^2^ = 0%, *p* = 0.75, Figure 4.

Conversion rate

We included thirteen trials [26,27,30,31,32,35,36,38,40,41,42,43,44] with 1220 patients that evaluated the conversion rate to open robotic platform liver patients. Our meta-analysis results demonstrated no significant difference between robotic and laparoscopic surgery in reducing the conversion rate for hepatectomy patients (OR = 0.84, 95% CI: 0.41–1.69, *p* = 0.62). The heterogeneity was observed with moderate certainty, I^2^ = 39%, *p* = 0.09, and can be seen in Figure 5.

Reoperation rate

Three trials [26,27,29] in Figure 6, including 366 patients, evaluated the reoperation rate of robotic surgery for hepatectomy patients. Our meta-analysis showed no significant difference between robotic and laparoscopic surgery in avoiding reoperation of hepatectomy patients (OR = 0.69, 95%CI: 0.25–1.90, *p* = 0.47). The heterogeneity was observed with low certainty, I^2^ = 0%, *p* = 0.63, Figure 6.

Blood loss

Figure 7, including 1754 patients in eighteen trials [26,27,29,30,31,32,33,34,35,36,37,38,39,40,41,42,43,44], evaluated the blood lossof robotic surgery for liver resection patients. Our meta-analysis results showed no significant difference between robotic and laparoscopic surgery in decreasing the amount of blood loss during surgery for hepatectomy patients (MD = −20, 95% CI: −64.90–23.34, *p* = 0.36). The heterogeneity was observed with higher certainly, I^2^ = 84%, *p* < 0.00001.

Operation time

Nineteen trials [26,27,28,29,30,31,32,33,34,35,36,37,38,39,40,41,42,43,44] (Figure 8) including 1783 patients evaluated the estimated operation time of robotic and laparoscopic platforms for liver surgery patients. Our meta-analysis results demonstrated a significant difference between robotic and laparoscopic surgery in reducing the operation time for liver surgery patients (MD = 43.99, 95%CI: 23.45–64.53, *p* < 0.0001). The heterogeneity observed was higher certainly, I^2^ = 86%, *p* < 0.00001.

Length of hospital stay

All seventeen studies of these 1672 patients [26,27,28,29,30,31,32,33,34,35,36,37,39,40,41,42,43] (Figure 9) assessed the length of hospital robotic surgery for hepatectomy patients. Our meta-analysis results showed that there was no significant difference between RH (robotic) and LH (laparoscopic) surgery in terms of length of hospital stay; we used a random effect model (MD = 0.10, 95% CI: −0.38–0.58, *p* = 0.69). The heterogeneity among studies was significant, and high, I^2^ = 75%, *p* < 0.00001.

Tumor size

Seven hundred seventy-five patients were included in twelve studies [28,29,32,35,36,37,38,39,40,41,42,44] (Figure 10) on the outcome of tumor size for liver resection in robotic surgery and using a laparoscopic platform. No significance was observed for either robotic or laparoscopic technique (MD = 0.30, 95% CI: −0–0.60, *p* = 0.05. Heterogeneity was observed with higher certainly, I^2^ = 71%, *p* = 0.0007.

Subgroup analyses

In this comparison subgroup analysis of minor hepatectomy, no significance was established in the subgroup analysis of results between robotic and laparoscopic hepatectomy, as shown in Figure 11. In the subgroup of minor operation time, nine studies were included [28,30,31,36,37,38,40,42,44], and MD = 39.87, 95% CI: −1.70–81.44, *p* = 0.06. The heterogeneity observed was considerable, I^2^ = 93%, *p* =0.06. In addition, in the subgroup of minor hospital stay, seven studies were included [28,30,31,36,37,40,42], and MD= 0.11, 95% CI: −0.32–0.54, *p* = 0.62. Heterogeneity was observed with lower certainly, I^2^ = 7%, *p* = 0.37. Figure 12 shows no significant difference between the two groups.

In the subgroup analysis (major hepatectomy) on operative time and blood loss (Figure 13 and Figure 14), respectively, a total of six studies [26,30,33,34,39,43] were included to compare robotic and laparoscopic techniques, and no significance was found (MD = −7.08, 95% CI: −15.22–0.07, *p* = 0.09). Heterogeneity was observed with substantial certainly, I^2^ = 65%, *p* = 0.01. Moreover, the value of the blood loss was (MD= −8.17 with 95%, CI = (−16.38–0.04), *p* = 0.05 and there was low heterogeneity, I^2^ = 31%, *p* = 0.20.

Figure 15 shows major hepatectomy subgroup analysis of complication rate, analysed in four studies [26,30,33,43]. Heterogeneity was low, I^2^ = 0%, *p* = 0.20. OR = 0.71.95%, CI= (0.42–1.19), *p* = 0.47. There was no significant difference between robotic and laparoscopic complication rate.

Number of patients with malignant liver tumors

Eight trials [29,30,32,35,36,38,41,42], including 865 patients, evaluated the number of patients with malignant liver tumors. Our meta-analysis results were similar in both techniques, robotic and laparoscopic platform, as shown in Figure 16. (OR = 0.99, 95%CI: 0.60–1.64, *p* = 0.98). The heterogeneity was observed as low certainty, I^2^ = 55%, *p* = 0.03. 

Body mass index (BMI)

The comparison of BMI for the overall group between robotic and laparoscopic hepatectomy was insignificant in both techniques [26,27,28,30,33,35,36,39,41,42,43] (Figure 17), (MD = 0.79, 95%CI: 0.25–1.34, *p* = 0.005). The heterogeneity was observed with low certainty, I^2^ = 27%, *p* = 0.20.

A total of three studies [35,41,44] showed that all cases in the laparoscopic group were performed with a pure laparoscopic platform, without hybrid incision or hand assistance, and all robotic hepatectomy patients were operated on using a DaVinci robot with four arms. Tsung et al. [41], confirmed that 76% of patients were also operated on with pure laparoscopy, and only 41% of liver resection patients were operated on with hand-assisted laparoscopy. 

## 3. Discussion

Recently, the laparoscopic approach has been highlighted as a technical advancement. In recent studies [20,45,46], the laparoscopic method has been shown to promote an improvement in various technicalities such as shorter period of hospitalization, lower morbidity, lower requirement for analgesic drugs after surgery, less blood loss during operation, and lower blood transfusion rate, compared with traditional open resection, which involves individual dissection and ligation of biliary and vascular structures. Meanwhile, robotic surgery has focused on improving outcomes of surgical procedures; this approach is exciting and attractive due to the application of improved technology, better image characteristic, the smaller size of robotic systems, and easy set-up. Robotic surgery has existed since the 1990s and continues in use in different specializations, especially cardiac surgery, urology, etc. However, although previous studies on general surgery have identified cost-effectiveness and extensive operation times as the main significant challenges in robotic surgery [47,48,49], robotic laparoscopic hepatectomy still offers an advanced and improved treatment opportunity. On the other hand, radiology procedures such as hepatocyte-specific magnetic resonance imaging (MRI) also provide a very important parametric detection tool for enhanced and sensitive diagnosis of patients with early stage HCC symptoms such as Gadoxetic acid, and precursor lesions [50,51,52].

Robotic assisted surgery is an alternative, minimally-invasive procedure, which is an innovative form of surgery and adopted in different medical specializations such as urology, gynecology, and other specialties [48]. Compared with traditional laparoscopic techniques, robotic surgery presents some benefits, especially deep manipulation into the abdominal cavity for treatment of anastomosis, and facilitates the handling of complex surgical procedures. Therefore, minimally-invasive procedures are the best choice for the treatment of HCC, metastases and tumors of benign conditions [15,17].

Several systematic review meta-analyses have demonstrated that the laparoscopic platform for liver surgery leads to earlier recovery, shorter length of hospital stay, and reduces postoperative pain compared to open liver surgery [8,53,54]. The initial objective of our study was to assess the clinical efficacity of robot-assisted hemi-hepatectomy versus laparoscopic hemi-hepatectomy, which is one of the most complicated procedures using a robotic platform [55], but our effort was stalled by insufficient data. Only four trials that exceptionally involved laparoscopic hemi-hepatectomy and robotic hemi-hepatectomy were included in our study with a total of 193 and 204 patients, respectively [26,30,33,34]. However, we re-focused our study by assessing the effective outcomes of laparoscopic and robotic hepatectomy. 

A recent study with meta-analyses by Guan et al. [56] reported that the use of robotic and laparoscopic technologies is equally practical and effective in terms of oncologic outcomes; similarly, their study asserted that robotic liver surgery can lead to long operation time and their explanation was based on the major liver resection. However, their assertion could not be substantiated since they did not conduct a specific meta-analysis on major hepatectomy between robotic and laparoscopic procedures major liver resection. Therefore, in addition to the general laparoscopic hepatectomy meta-analysis, our study conducted a subgroup analysis between robotic and laparoscopic surgery on major liver resection. 

This meta-analysis results showed that the robotic procedure was related to longer operation time. In addition, this study found significant differences between robotic and laparoscopic in operation time. In the subgroup analysis of the major hepatectomy, three outcomes were included: operation time, estimated blood loss and complication rate, however, no significant difference was observed between laparoscopic and robotic liver resection. From this evidence, our meta-analysis results showed that major liver resection is far away to lead a long operation in robotic hepatectomy, more experience from surgeon could decrease the operation time. Additionally, our subgroup analysis comparing minor hepatectomy using robotic and laparoscopic liver surgery the operation time and hospital stay showed no significant difference wherever a higher heterogeneity was generated.

However, there was more conversion laparoscopic groups compared to robotic groups and uncontrolled bleeding may lead to open conversion to the robotic hepatectomy [27,57]. Tsung reported the outcome of conversion rate to open surgical procedure was similar in robotic contrasted with laparoscopic groups [32]. A comparative study [35], found a significantly higher conversion rate to open surgical procedures in RH compared to LH with (20% vs. 7.6%, *p* = 0.034), respectively. Based on our knowledge, there were no significant differences between the two groups robotic and laparoscopic hepatectomy in conversion rate and the oncological outcomes number of tumor and tumor size, the results were similar in both techniques. And was no important difference between laparoscopic hepatectomy and robotic hepatectomy in the length of stay, and also in the estimated blood loss. On the other hand, complications during surgery could lead to long operations and more blood loss and may raise the hospital stay because patients need more time to recover [58,59,60,61].

In this study, no difference was introduced for the complication rate. Different surgeon levels for hepatic resection and patient morbidity may be conducted [62]. A recent retrospective study based on left hemihepatectomy confirmed that RH was related to decreased intraoperative blood loss compared to LH and no significant difference in operation time [26]. According to our result with significantly higher heterogeneity in blood loss, operation time and length of hospital, difficult to accomplish due to several resection levels. 

In terms of clinical comprehensiveness of real or potential cases, our work was limited in that we did not include any randomized control trial because it did not meet our inclusion criteria. If such is of particular interest to a physician in search of relevant decision, this aspect may be assessed based on other reports. Also, beyond the specific scope of this review was the cost of both techniques, which was not evaluated. For consideration of cost analysis for establishment of new health facilities or upward equipment of existing facilities, cost information is not covered here. For fewness of items documenting pure hemihepatectomy using robotic or laparoscopic methods; we could not pursue a meta-analysis. Further methodology in scientific research needs to achieve those limitations

## 4. Conclusions

The study results show that the outcome of operation time was significant, and robotics lead to extended operation time. No significant differences were observed between the two groups, robotic and laparoscopic, in blood transfusion rate, blood loss, conversion rate, length of hospital stay, and incidence of reoperation. Additionally, the subgroup analysis for major and minor robotic and laparoscopic liver resection were also not significantly different. Therefore, scientific evaluation research focusing on a specific portion of the liver may be better for more efficacity and precise results. More randomized study needs to be conducted to evaluate this field.

## Figures and Tables

**Figure 1 jcm-11-05831-f001:**
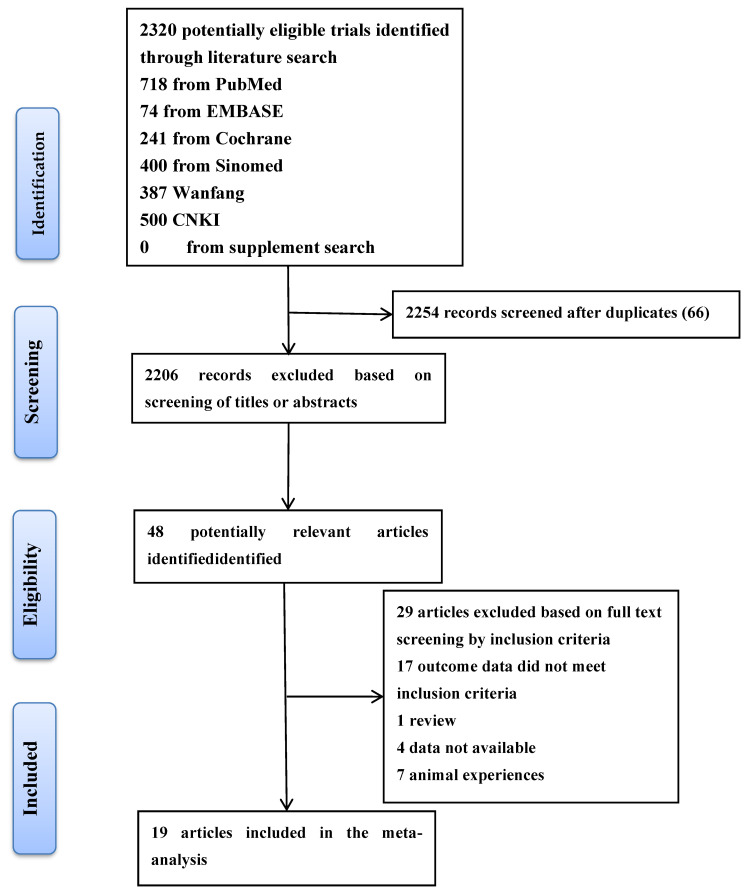
Flow diagram of the literature screening process and results.

**Figure 2 jcm-11-05831-f002:**
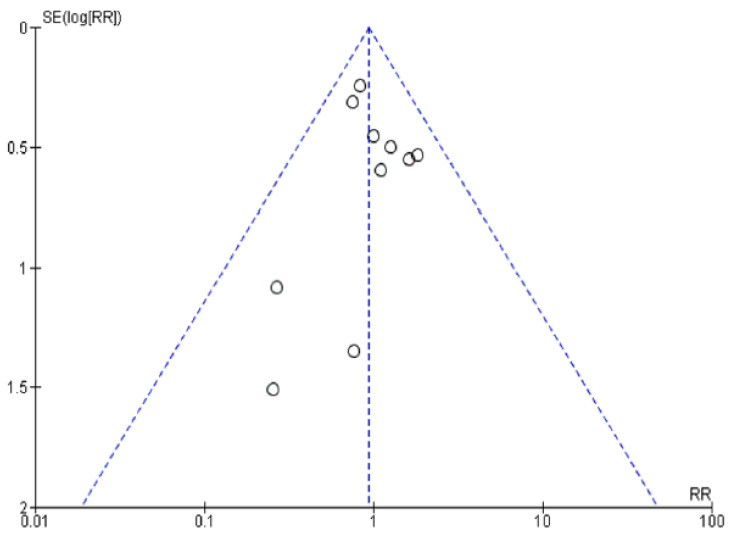
Funnel plot of complication.

**Figure 3 jcm-11-05831-f003:**
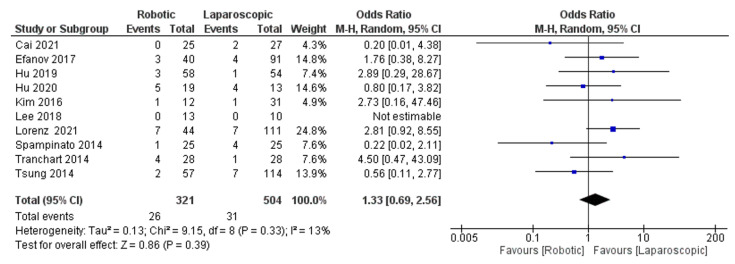
Forest plot of blood transfusion [26,29,31,32,34,36,37,39,42,43].

**Figure 4 jcm-11-05831-f004:**
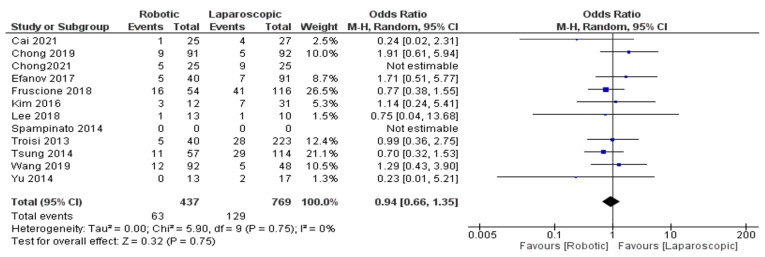
Forest plot of complication rate [26,27,29,30,32,33,35,36,37,40,43].

**Figure 5 jcm-11-05831-f005:**
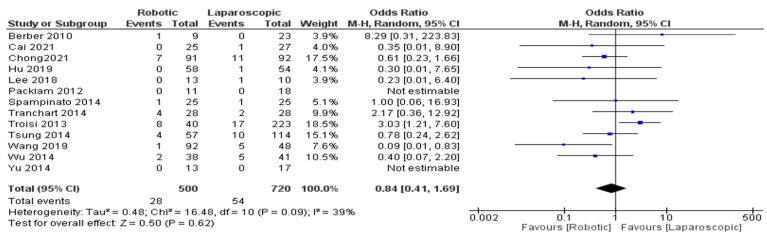
Forest plot of conversion to open [26,27,30,31,32,35,36,38,40,41,42,43,44].

**Figure 6 jcm-11-05831-f006:**
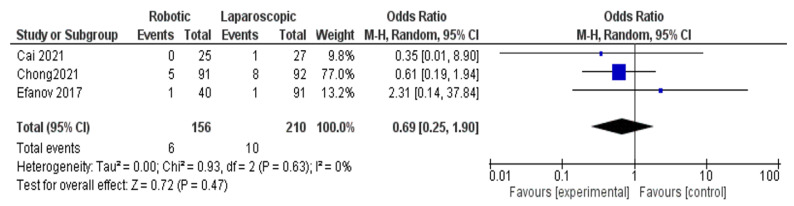
Forest plot of reoperation rate [26,27,29].

**Figure 7 jcm-11-05831-f007:**
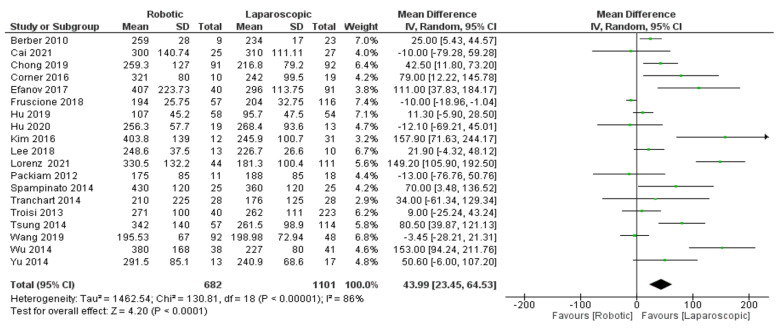
Forest plot of estimated Blood loss [26,27,29,30,31,32,33,34,35,36,37,38,39,40,41,42,43,44].

**Figure 8 jcm-11-05831-f008:**
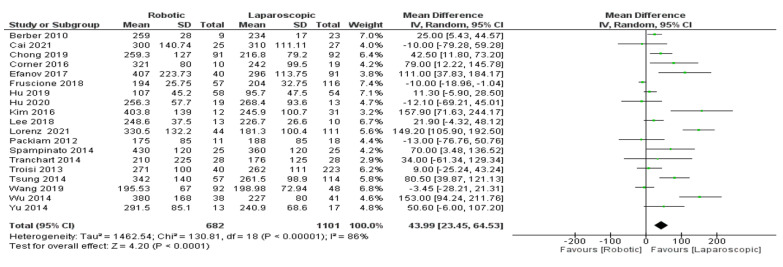
Forest plot of operation time [26,27,28,29,30,31,32,33,34,35,36,37,38,39,40,41,42,43,44].

**Figure 9 jcm-11-05831-f009:**
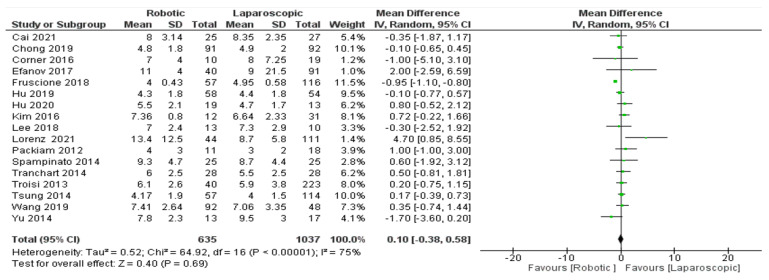
Forest plot of length of hospital (day) [26,27,28,29,30,31,32,33,34,35,36,37,39,40,41,42,43].

**Figure 10 jcm-11-05831-f010:**
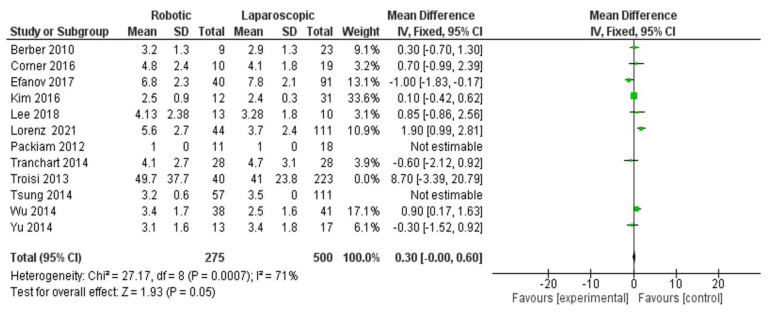
Forest plot of tumor size for liver resection [28,29,32,35,36,37,38,39,40,41,42,44].

**Figure 11 jcm-11-05831-f011:**
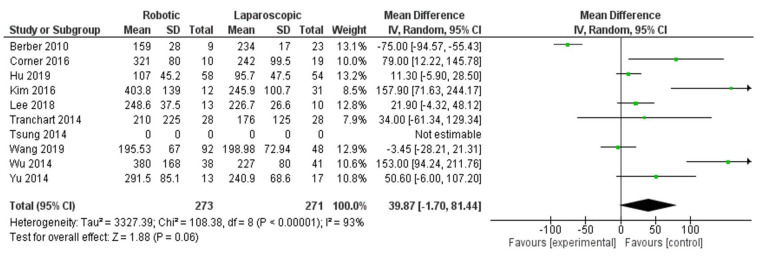
Forest plot subgroup of minor operation time [28,30,31,36,37,38,40,42,44].

**Figure 12 jcm-11-05831-f012:**
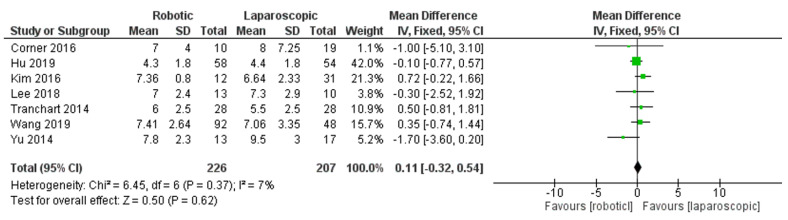
Forest plot subgroup of minor hospital stay [28,30,31,36,37,40,42].

**Figure 13 jcm-11-05831-f013:**
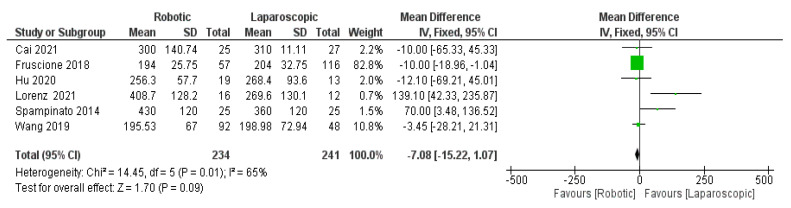
Forest plot of subgroup analysis of major on operation time [26,30,33,34,39,43].

**Figure 14 jcm-11-05831-f014:**
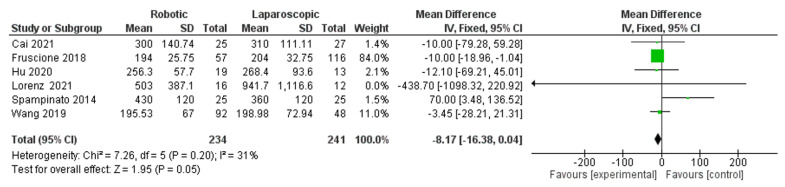
Forest plot of subgroup analysis of major blood loss [26,30,33,34,39,43].

**Figure 15 jcm-11-05831-f015:**
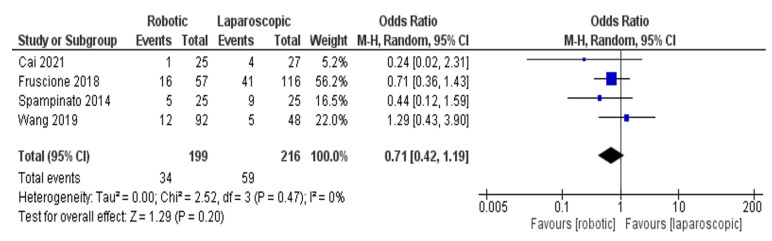
Forest plot of Subgroup analysis of complication rates [26,30,33,43].

**Figure 16 jcm-11-05831-f016:**
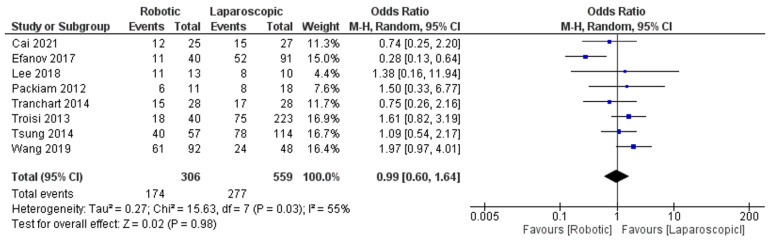
Number of patients with malignant liver tumors [20,29,32,35,36,38,41,42].

**Figure 17 jcm-11-05831-f017:**
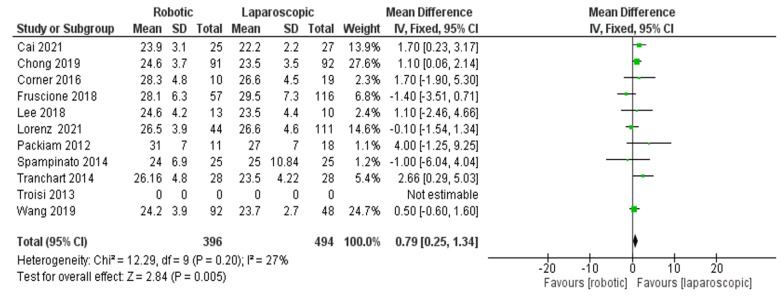
BMI for the overall group [26,27,28,30,33,35,36,39,41,42,43].

**Table 1 jcm-11-05831-t001:** Characteristics of included studies in the meta-analysis.

Author	Country	Period	Design	Group	Total	Sex *n* (M%)	Mean Age	BMI (kg/m^2^)	Tumor Size(cm) mm	Pure Hemi Hepatectomy, *n* (%)
Cai [26], 2021	China	2015–2020	Retrospectives	RH	25	12 (48.0%)	56.4 ± 9.1	23.9 ± 3.1	5.5 ± 2.3	YES
				LH	27	18 (66.7%	52.7 ± 11.6	22.2 ± 2.2	4.3 ± 1.9	
Chong [27], 2019	China	2003–2017	Prospective	RH	91	NA	58.7 ± 11.7	24.6 ± 3.7	<3 cm, ≥3 cm	NO
				LH	92	NA	59.8 ± 11.9	23.5 ± 3.5	<3 cm, ≥3 cm	NO
Croner [28], 2016	Germany	2011–2015	Retrospectives	RH	10	NA	64 (45–76	28 (28.3)	4.8 (2.9–10.5	NO
				LH	19	NA	59 (32–85)	27 (26.6)	4.1 (1.8–8.5)	NO
Efanov [29], 2017	Russia	2010–2016	Retrospectives	RH	40	9 (NA)	45(18–76)	NA	73 (17–142)	NO
				LH	91	36(NA)	51(21–77)	NA	64 (8–180)	NO
Wang [30], 2019	China	2011–2017	Retrospectives	RH	92	55 (59.8)	54.1 ± 11.2	24.2 ± 3.9	<5 cm, ≥5 cm	YES
				LH	48	24 (50.0)	49.4 ± 13.0	23.7 ± 2.7	<5 cm, ≥5 cm	
Hu [31], 2019	China	2015–2017	Retrospective	RL	58	33	52.2 years	24.7	NA	NO
				LH	54	26	48.9 years	23.8	NA	NO
Tsung [32], 2014	USA	2007–2011	Retrospective	RH	57	24 (42%)	58.35 ± 14.6	NA	3.15 (2.05–5.00)	NO
				LH	114	47 (41%)	58.72 ± 15.8	NA	3.50 (2.0–6.0)	NO
Fruscione [33], 2018	USA	2011–2016	Retrospective	RH	57	20 (35.1)	58.1 (15.7)	28.1 (6.3)	NA	YES
				LH	116	52 (44.8%)	53.2 (15.4)	29.5 (7.3)	NA	
Hu [34], 2020	China	2011–2017	Retrospectively	RH	19	(10.5%)	49.2 ± 10.6	1.6 ± 0.2	>10 cm	YES
				LH	13	(7.7%)	46.5 ± 8.9	1.6 ± 0.2	>10 cm	
Troisi [35], 2013	Belgium-Italy	2004–2010	Retrospective-comparative	RH	40	(67.5%)	64.6 ± 12.1	NA	51.8 ± 37.6 (1–19)	NO
				LH	233	43.9%	55.3 ± 15.7	NA	49.7 ± 37.7 (1–20)	NO
Lee [36], 2018	South Korea	2016–2018	Retrospective	RH	13	NA	62.2 ± 9.	24.6 ± 4.2	41.3 ± 23.8	NO
				LH	10	NA	58.8 ± 11.2	23.5 ± 4.4	32.8 ± 18.0	NO
Kim [37], 2016	South Korea	2007–2013	Retrospective	RH	12	6 (50%)	54.1 ± 12.2	NA	2.3 (2.0–3.6)	NO
				LH	31	18 (58%)	56.4 ± 11.6	NA	2.4 (1.7–3.0)	
Berber [38], 2010	USA	2008–2009	Prospective	RH	9	7 (77.8%)	66.6 ± 6.4	NA	3.2 ± 1.3	NO
				LH	23	12 (52%)	66.7 ± 9.6	NA	2.9 ± 1.3	NO
Lorenz [39], 2021	Germany	2010–2020	Retrospective	RH	44	24 (54.5%)	62.6 ± 14.5	26.5 ± 3.9	5.6 ± 2.7	
				LH	111	50 (45.0%)	61 (55.0)	27.0 ± 4.6	3.7 ± 2.4	
Yu [40], 2014	South Korea	2007–2011	Case Control	RH	13	7 (53.9%)	50.4 ± 12.2	NA	31.1 ± 16.0	NO
				LH	17	9 (52.94%)	52.5 ± 9.7	NA	34.8 ± 18.2	NO
Packiam [41], 2012	USA	2009–2011	Retrospective	RH	11	3 (27%)	57 ± 16	31 ± 7	5.5(2.4–6.5)	NO
				LH	18	4 (22%)	52 ± 17	29 ± 7	4.4 (2.6–7.1)	NO
Tranchart [42], 2014	France-Italy	2008–2013	Matched design	RH	28	13 (46.4%)	66.5 (42–84)	26.1 (16.7–36)	35(6–115)	NO
				LH	28	13 (46.4%)	66(41–78)	23.2 (16–33)	40(6–130)	NO
Spampinato [43], 2014	Italy	2009–2012	Retrospective	RH	25	13 (52%)	63 (32–80)	24 (16.4–21.8)	NA	NO
				LH	25	10 (40%)	62 (33–80)	25 (20–28.5)	NA	NO
Wu [44], 2014	Taiwan	2007–2011	Retrospective	RH	38	32 (84.2%)	60.9 ± 14.9	NA	3.4 ± 1.7	NO
				LH	41	28 (68.3%)	54.1 ± 14	NA	2.5 ± 1.6	

NA: not available, LH: Laparoscopic hepatectomy, RH: Robotic hepatectomy.

**Table 2 jcm-11-05831-t002:** Characteristics of included studies in the meta-analysis (continued).

Author	Group	Surgical Duration (minute)	Complications	Blood Loss (mL)	Conversion to Open	Transfusion	Reoperation	Length of Hospital (day)	Hepatectomy Extend
Cai [26], 2021	RH	303.578 ± 149.3624	1/25	100 ± 37.037	0/25	0/25	0/27	8 ± 3.1445	left hemihepatectomy: 25(48.07%)
	LH	313.57 ± 117.40	4/27	200 ± 148	1/27	6/27	1/27	8.35 ± 2.34	left hemihepatectomy: 27(51.92%)
Chong [27], 2019	RH	259.3 ± 127.0	9/91	274.6 ± 568.1	7/91	NA	5/91	4.8 ± 1.8	LLS:39 (42.9%) Wedge resection:31 (34.1%) Left hepatectomy:39 (42.9%) Right hepatectomy:6 (6.6%) CLR: 1 (1.1% MR: 2 (2.2%) Major:19 (20.9%) Minor: 72 (79.1%)
	LH	216.8 ± 79.2	5/92	212.4 ± 313.4	11/92	NA	8/92	4.9 ± 2.0	LLS: 40 (43.5%) Wedge resection:47 (51.1%) Left hepatectomy: 7 (51.1%) Right hepatectomy: 1 (1.1%) CLR:0 MR:0 Major: 4 (4.3%) Minor: 88 (95.7%)
Croner [28], 2016	RH	321 ± 80	NA	306 mL NA	NA	NA	NA	7 ± 4	Minor: 10 (34.48%)
	LH	242 ± 99.5	NA	356 mL (NA)	NA	NA	NA	8 ± 7.25	Minor: 19 (65.515)
Efanov [29], 2017	RH	407 ± 223.73	5/40	465 ± 500	NA	3/40	1/40	11 ± 4	RH-H:0 LH-H: 2 (5%) RPS: 5 (13%) S: 1 (3%) WRPS: 3 (8%) ALS-S: 18 (45%) WRAS: 11 (28%)
	LH	296 ± 133.75	7/91	302 ± 550	NA	4/91	1/91	9 ± 21.5	RH-H: 9 (10%) LH-H: 2 (2%) RPS: 6 (7%) S: 6 (7%) WRPS: 23 (25%) ALS-S: 24 (26%) WRS: 21 (23%)
Wang [30], 2019	RH	195.53 ± 67.00	12/92	346.04 ± 234.17	NA	NA	NA	7.41 ± 2.64	Left liver: 48 (52.2%) Right liver: 44 (47.8%)
	LH	198.98 ± 72.94	5/48	243.04 ± 171.87	NA	NA	NA	7.06 ± 3.35	Left liver: 29 (60.4%) Right liver: 19 (21.6%)
Hu [31], 2019	RL	107.0 ± 45.2	NA	80.1 ± 144.4	0/58	NA	NA	4.3 ± 1.8	Left lateral sectionectomy:51.17 %
	LH	95.7 ± 47.5	NA	108.9 ± 180.8	1/54	NA	NA	4.4 ± 1.8	Left lateral sectionectomy:48.21%
Tsung [32],2014	RH	353.66 ± 143.75	11/57	195.58 ± 218.66	4/57	2\57	NA	4.1767± 1.90	Major:21(NA) Minor:36(NA)
	LH	261.5 ± 98.9	29/114	170.34 ± 225.25	10/114	7\114	NA	4 ± 1.50	Major:42(NA) Minor:72(NA)
Fruscione [33], 2018	RH	195.537 ± 22.457	16/57	268.18 ± 103.56	NA	NA	NA	4 ± 0.4361	Left: 20 (35.1) Patial: 17 (29.8) Right: 20 (35.1)
	LH	205.0674 ± 25.6799	41/116	405.08 ± 117.62	NA	NA	NA	4.9492 ± 0.5881	Left: 22 (19.0) Patial: 48 (41.4) Right: 46 (39.7)
Hu [34], 2020	RH	268.4 ± 93.6	NA	319.5 ± 206.0	NA	5\19	NA	5.5 ± 2.1	Right:15 Left:4
	LH	268.4 ± 93.6	NA	476.9 ± 210.8	NA	4\13	NA	4.7 ± 1.7	Right:8 Left:5
Troisi [35], 2013	RH	271 ± 100	5/40	NA	8/40	NA	NA	6.1 ± 2.6	Major hepatectomy:0 Left hepatectomy:0 Right hepatectomy:0
	LH	262 ± 111	28/223	NA	17/223	NA	NA	5.9 ± 3.8	Major hepatectomy: 37 (16.6%) Left hepatectomy:16 (7.2%) Right hepatectomy:17 (7.6%)’& other extents
Lee [36], 2018	RH	248.6 ± 37.5	NA	320.3 ± 331.9	0/13	0/13	NA	7.0 ± 2.4	left-sidehepatectomy Left lateral sectionectomy
	LH	226.7 ± 26.6	NA	392.8 ± 374.5	1/10	0/10	NA	7.3 ± 2.9	
Kim [37],2016	RH	403.8 ± 139.0	NA	206.6875 ± 125.79	NA	1/12	NA	7.36 ± 0.8386	left lateral sectionectomy: (27.90%)
	LH	245.9 ± 100.7	NA	212.3508 ± 291.4453	NA	1/31	NA	6.6437 ± 2.3316	left lateral sectionectomy: (72.09%)
Berber [38], 2010	RH	258.5 ± 27.9	NA	136 ± 61	1/9	NA	NA	NA	Segmental liver resection:6 Left lateral sectionectomy:3
	LH	233.6 ± 16.4	NA	155 ± 54	0/23	NA	NA	NA	Segmental liver resection:12 Left lateral sectionectomy:11
Lorenz [39], 2021	RH	330.5 ± 132.2	4/44	439.8 ± 346.3	NA	7/44	NA	13.4 ± 12.5	Major: 16, Minor: 25
	LH	181.3 ± 100.4	3/111	425.4 ± 590.1	NA	7/111	NA	8.7 ± 5.8	Major: 12, Minor: 60
Yu [40], 2014	RH	291.5 ± 85.1	0/13	388.5 ± 65.0	0/13	0/13	NA	7.8 ± 2.3	LLS:10, LH-H:3
	LH	240.9 ± 68.6	2/17	342.6 ± 84.7	0/17	0/17	NA	9.5 ± 3.0	LLS:6, LH-H:11
Packiam [41], 2012	RH	175 ± 85	—	30 ± 40	0/11	0	NA	4 ± 3	LLS
	LH	188 ± 85	—	30 ± 35	0/18	0	NA	3 ± 2	LLS
Tranchart [42], 2014	RH	210 ± 125	NA	200 ± 150	4/28	4/28	NA	6 ± 2.5	Bisegmentectomy:1,LLS:5, Segmentectomy:7, etc..
	LH	176 ± 125	NA	150 ± 150	2/28	1/28	NA	5.5 ± 2.5	Bisegmentectomy:1,LLS:5, Segmentectomy:7, etc..
Spampinato [43], 2014	RH	430 ± 120	5/25	250 ± 155	1/25	1/25	NA	9.3 ± 4.7	Major hepatectomy
	LH	360 ± 120	9/25	400 ± 155	1/25	4/25	NA	8.7 ± 4.4	Major hepatectomy
Wu [44], 2014	RH	380 ± 166	NA	325 ± 480	2/38	NA	NA	—	Major liver resection
	LH	227 ± 80	NA	173 ± 165	5/41	NA	NA	—	Right &Left lobe

NA: not available, LH: Laparoscopic hepatectomy, RH: Robotic hepatectomy, CLR: Caudate lobe resection, MR: Multiple resections, LLS: Left lateral sectionectomy, RH-H: Right hemi-hepatectomy, LH-H: Left hemi-hepatectomy, RPS: Right posterior sectionectomy, S: Segmentectomy, WRPS: Wedge resection of posterosuperior (1,4a,7,8) segment, ALS-S: Anterolateral segment/sectionectomy, WRAS: Wedge resection of anterolateral segments.

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
