# Peer review of "Laparoscopic versus Robotic Hepatectomy: A Systematic Review and Meta-Analysis"

_jcm, 2022, doi:10.3390/jcm11195831_

Round 1

Reviewer 1 Report

I read carefully the paper, but I think it should be improved:

- The introduction needs to be revised focusing in the results already achieved by others

- The discussion is superficial, I think you should better analyze pitfalls and drawbacks of the other study, compare better the results of the authors

- The review analysis could be better including more article with the aim to achieve more robust results

Author Response

We appreciate the reviewer very much for offering us such careful suggestion.

Reviewer 2 Report

In this systematic review, the authors addressed the critical topic of Laparoscopic and Robotic Hepatectomy, analyzing twelve studies, and elucidated the outcomes of robotic and laparoscopy with 489 and 838 patients, respectively.

The manuscript is well organized and presented. However, in my opinion, in a study addressing Laparoscopic and Robotic Hepatectomy outcomes, two important topics should be recalled in the discussion.

-Both Laparoscopic and Robotic Hepatectomy may present significant advantage in patients with subottimal liver function. I would suggest to discuss the role of Laparoscopic and Robotic Hepatectomy in patients with Child-Pugh class B patients as they are not good candidate to traditional surgery, as recently discussed in a comprehensive review (Non-transplant therapies for patients with hepatocellular carcinoma and Child-Pugh-Turcotte class B cirrhosis. Lancet Oncol. 2017 Feb;18(2):e101-e112. doi: 10.1016/S1470-2045(16)30569-1).

-Of relevance, Laparoscopic and Robotic Hepatectomy may be the best therapeutic option for patients with early diagnosis of early stage of HCC. In this regard, the authors should recall the role of imaging and in particular the magnetic resonance with hepatobiliary contrast agents such as gadoxetic acid, because it has been demonstrated that such a imaging technique may allow the diagnosis of early HCC also when the typical arterial heyperenhancement lacks, as previously reported (Impact of gadoxetic acid (Gd-EOB-DTPA)-enhanced magnetic resonance on the non-invasive diagnosis of small hepatocellular carcinoma: a prospective study. Aliment Pharmacol Ther. 2013 Feb;37(3):355-63. doi: 10.1111/apt.12166.).

Author Response

We appreciate the reviewer for drawing our attention to this aspect

Reviewer 3 Report

This is a well written paper on an interesting topic. However, the results on robotic hepatectomy vs laparoscopic hepatectomy in terms of blood transfusion rate, conversion rate, estimated blood loss, length of hospital and incidence of reoperation seem insufficient. The work may be improved by providing additional comparisons, such as operative time, tumor size and number, and BMI for the overall group, and also comparisons between subgroups, such as minor vs major hepatectomy, pure vs assisted hepatectomy.

Author Response

(The authors gave the same response as above.)

Round 2

Reviewer 3 Report

The paper was improved after revision and it should be revised:

- English proofing

- Abstract revision, according to Background, Methods, Results, Conclusions

Author Response

The paper was improved after revision and it should be revised:

- English proofing

Response :We appreciate the reviewer very much for offering us such careful suggestion.

 The English has been proofed.

- Abstract revision, according to Background, Methods, Results, Conclusions

Response: Thank you very much for the reviewer suggestion

This study aimed to assess the surgical outcomes of robotic compared to laparoscopic hepatectomy, with a special focus on the meta-analysis method. Original studies were collected from Chinese databases PubMed, EMBASE, and Cochrane Library databases. Our systematic review was conducted on 682 patients with robotic liver resection, and 1101 patients performed laparoscopic platform. Robotic surgery has a long surgical duration and more blood loss compared to laparoscopic surgery (MD=43.99, 95% CI: 23.45-64.53,P=0.0001), and (MD=43.99, 95%CI: 23.45-64.53, P <0.0001, respectively, while no significant difference in length of hospital (MD=0.10, 95% CI: -0.38-0.58, P=0.69),  the incidence of  conversion to open (OR=0.84, 95% CI: 0.41-1.69, P=0.62),  tumor size (MD=0.30, 95% CI: -0-0.60, P=0.05), as well  as the subgroup analysis major and minor hepatectomy on  operation time (MD= -7.08, 95% CI: -15.22-0.07, P=0.09) and( MD=39.87, 95% CI: -1.70-81.44, P=0.06), respectively. However, deficiency of robotic hepatectomy for extended operation time and more blood loss compared to laparoscopic, robotic hepatectomy is still effective and equivalent to laparoscopic hepatectomy in their outcomes. Scientific evaluation research on one portion of the liver may be better for more efficacity and precise results. Therefore more clinical trials need to evaluate the clinical outcomes of robotic compared to laparoscopic hepatectomy.
